

# Risk factors for violent crime in patients with schizophrenia: a retrospective study

Ruoheng Lin[1], Qiguang Li[2], Ziwei Liu[3], Shaoling Zhong[4], Ying Huang[1], Hui Cao[5], Xiangbin Zhang[6], Jiansong Zhou[1] and Xiaoping Wang[1]

[1] Department of Psychiatry, National Clinical Research Center for Mental Disorders, and National Center for Mental Disorders, The Second Xiangya Hospital of Central South University, Changsha, China
[2] Xi'an Mental Health Center, Xi'an, China
[3] Department of Preventive Medicine, School of Medicine, Hunan Normal University, Changsha, China
[4] Department of Community Mental Health, the Affiliated Brain Hospital of Guangzhou Medical University, Guangzhou, China
[5] Department of Psychiatry, Brain Hospital of Hunan Province, The Second People's Hospital of Hunan Province, Changsha, China
[6] Department of Neurology, Xiangya Hospital, Central South University, Changsha, China

Corresponding authors
Jiansong Zhou, zhoujs2003@126.com, zhoujs2003@csu.edu.cn
Xiaoping Wang, xiaop6@csu.edu.cn

## ABSTRACT

**Introduction**. The relationship between schizophrenia and violence is heterogeneous and complex. The aim of this study was to explore the characteristics and the potential risk factors for violence crime in patients with schizophrenia.

**Methodology**. We conducted a retrospective case-control study at the Judicial Psychiatric Identification Unit of Xiangya Second Hospital of Central South University from January 1, 2013 to December 31, 2016. The case group included violent offenders diagnosed with schizophrenia, while the control group comprised non-violent individuals with the same diagnosis.

**Results**. There were 308 individuals in the violent group [subdivided into the homicide group ($n = 155$) and the intentional injury group ($n = 153$)] and 139 individuals in the non-violent group. A risk model showed that a history of violence (odds ratio (OR) = 2.88, 95% CI [1.79–4.64]), persecutory delusions (OR = 2.57, 95% CI [1.63–4.06]), regular treatment in the previous four weeks (OR = 0.29, 95% CI [0.16–0.51]) and insight (OR = 0.30, 95% CI [0.14-0.62]) were independently associated with violence.

**Conclusion**. This study provided useful clinical information to identify risk factors for violence and develop better strategic programs to manage violence in patients with schizophrenia.

## INTRODUCTION

Previous studies have found that people with schizophrenia are at significantly higher risk for violence than the general population (*Fazel et al., 2009b*; *Fazel et al., 2009a*; *Short et al., 2013*), and these studies have had several adverse effects, including discrimination, stigma, and negative effects on treatment outcomes (*Yanos et al., 2020*; *Barlati et al., 2022*). However, in recent years, a number of scholars have taken the opposite view, arguing that people with serious mental disorders are more often victims of violence than perpetrators

(*Latalova, Kamaradova & Prasko, 2014*; *Rossa-Roccor, Schmid & Steinert, 2020*; *Fazel & Sariaslan, 2021*). Regardless of perspective, the lethality of violent behavior by people with schizophrenia requires that mental health providers identify and intervene in a timely manner with correlates of violent behavior to minimize the risk of violence. Violent behavior in patients with schizophrenia is associated with a variety of factors such as sociodemographic, therapeutic, psychopathological factors (*Bo et al., 2011*), and biological factors (*e.g.*, cognitive deficits) (*Reinharth et al., 2014*; *Barlati et al., 2023*). However, studies vary greatly in the way they define violence; For example, some studies include threats and verbal aggression when defining the term violence (*Douglas, Guy & Hart, 2009*), while some others define violence as physical aggression (*Bulgari et al., 2017*). There was a study that classifies acts of attacking staff member, other patients, or damaging property within the category of violence (*Grassi et al., 2001*). The Chinese legal system defines violence against others as acts that target the person or property, cause significant damage to the victim's physical or mental health, life, or property, and directly endanger his or her life and health (*Yang, Kang & Yang, 2001*). These differences may be one of the reasons for the inconsistent findings on factors associated with violent behavior in schizophrenia. Some studies have reported an association between positive symptoms of schizophrenia and violent behavior (*Witt, Van Dorn & Fazel, 2013*; *Zhou et al., 2016*; *Brucato et al., 2018*; *Li et al., 2019*). For instance, drawing from our team's previous research, positive symptoms, such as current delusions and excitement, were found to be correlated with aggression in general clinical settings in China. However, contrasting results have been observed by others, as they did not find a similar correlation (*Marshall et al., 2016*; *Coid et al., 2018*; *Buchanan et al., 2019*). Therefore, when studying factors associated with violence in patients with schizophrenia, it is important to choose a more appropriate method to assess violence. Due to the high correlation between violent crime and mental disorders (*Fleischman et al., 2014*), the subjects of the present study were patients with schizophrenia who committed violent crimes, and violation of law was used as a measure for violence.

China is a developing country with a large population; thus, the unique population characteristics may produce violence-related factors that are different from those in Western countries among patients with schizophrenia. It has been found that comorbid drug use disorders are strongly associated with a higher risk of violence among patients with schizophrenia in some Western countries (*Fazel et al., 2014a*). However, epidemiological studies from China offer a different perspective. In 2019, Huang and colleagues conducted a large-scale epidemiological study in China, finding that the lifetime prevalence rate of drug use disorders in the Chinese population is approximately 0.4% (*Huang et al., 2019*). In contrast, studies in the United States indicate that the lifetime prevalence rate of drug use disorders in the American population is about 10% (*Ignaszewski, 2021*). This may suggest that the risk of violence among people with schizophrenia in China may be influenced by other factors than just drug use disorders. It is also suggested that, for the assessment of the risk of violence among patients with schizophrenia, more attention should be given to specific psychotic symptom rather than the general disease state (*Swanson et al., 2006*; *Douglas, Guy & Hart, 2009*). For instance, *Swanson et al. (2006)* found that "positive" psychotic symptoms (*e.g.*, persecutory ideation) increased the risk

violence, while "negative" psychotic symptoms (*e.g.*, social withdrawal) lowered the risk of violence. Since psychotic symptoms (especially delusions) may be culturally relevant and differ across cultures, for example, *Zheng (1989)* found significant differences between China and Japan in the content of exaggerated delusions, descent delusion, victimization delusions, and possession delusions, the relationship between psychotic symptoms and violence in Chinese patients with schizophrenia may also have characteristics that differ from those in Western countries.

Homicidal behavior, as a more severe form of violence, is not commonly observed among patients with schizophrenia. For example, a study in the UK found that only 6% of homicides were committed by individuals with schizophrenia (*Appleby, Shaw & Amos, 1997*). However, due to the lethal nature of homicidal behavior in patients with schizophrenia, it often has a profound impact on the families and friends of the victims (*Hodgins, 2008*). Therefore, it is necessary to analyze the characteristics of homicidal behavior separately.

In the present study, we aimed to explore the relationship between factors based on the Chinese population characteristics and violent crime among patients with schizophrenia, construct a model of violence risk assessment, and provide an accurate approach to violence prevention among Chinese patients with schizophrenia.

# MATERIALS & METHODS

## Study design and participants

The methodology employed in this study is based on the approach detailed in our preprint (*Lin, Li & Liu, 2023*). From January 1, 2013, to December 31, 2016, we retrospectively analyzed archival data and medical records of patients with schizophrenia who received forensic psychiatric assessments in the Judicial Psychiatric Appraisal Unit of the Second Xiangya Hospital of Central South University. We planned to conduct multiple regression analysis to examine the factors related to violent crimes by patients with schizophrenia, and the required number of subjects is ten times the number of independent variables (*Okamura et al., 2021*). Since there were approximately 25 independent variables in this study, the sample size needed to be 250, and considering a 10% refusal rate and potential data loss, we expanded the final sample size to 275 people. Based on the presence or absence of violent crimes, patients were categorized into the case group and the control group. Within the case group, further subgroups were formed based on the severity of violence, including homicide and intentional injury categories. The inclusion criteria were (1) aged 18–60 years, (2) diagnosed with schizophrenia according to the International Classification of Diseases criteria, 10th edition (ICD-10) (*Geneva*), which was the specific criteria used for the diagnosis, based on a clinical diagnostic interview conducted by at least two qualified forensic psychiatrists, (3) with normal intelligence. The exclusion criteria were (1) patients with incomplete socio-demographic data in the case file, (2) patients who were uncooperative in the mental status assessment, (3) patients with hearing loss or verbal communication difficulties that precluded them from completing the assessment, and (4) patients with comorbidities of other severe mental disorders (such as mental

retardation, dementia, epilepsy and severe traumatic brain injury, *etc.*). Data collected from each patient's file data and medical records were recoded to remove any identifiable information (*e.g.*, name or date of birth). All methods were performed in accordance with relevant guidelines and regulations. All participants provided informed consent. The study was approved by the Ethics Committee of the Second Xiangya Hospital (2013068).

## Definition of violence

In the present study, violent crimes refer to crimes involving serious acts of violence, including homicide, aggravated assault, robbery, sexual coercion, and rape. Nonviolent crimes involve other relatively benign offenses such as smuggling, drug trafficking, theft, fraud, and racketeering. To explore the potential differences in influencing factors between violence of different severities, the violence group was further divided into the homicide group and the intentional injury group. The homicide group included those who had committed murder, manslaughter and infanticide.

## Tools and assessments

A self-designed standardized form was used to record the basic demographic and clinical information of all subjects. Demographic data included age, gender, level of education, marital status, household registration, ethnicity, and work status; the following clinical data were recorded: duration of illness, duration of untreated psychosis, previous treatment, regular treatment in the previous four weeks, family history of mental disorders, history of drug use, alcohol abuse, history of violence, and history of self-harm. Based on a prior study (*Witt, Van Dorn & Fazel, 2013*), we assigned codes to specific symptoms that have been found to be associated with violence in psychosis. These symptoms include command hallucinations, persecutory delusions, delusions of jealousy, depression, excitement, apathy, and insight (*Witt, Van Dorn & Fazel, 2013*). By reviewing forensic files or assessment records, the researchers identified whether patients exhibited any of the above psychotic symptoms within days to a week before the index offense. Each symptom was rated 0 (not present) or 1 (present).

## Statistical analysis

All statistical analyses were performed using the IBM SPSS 25 software. The Shapiro–Wilk test was used to check the distribution of continuous variables. Characteristics of all participants were summarized as frequency (percentage) for categorical variables and median and interquartile range (IQR) for continuous variables. For the comparison of categorical variables, the chi-square test or Fisher's exact test was used. Given the non-normal distribution of continuous variables, the comparison between two groups was conducted using the Mann–Whitney $U$ test, and the comparison among three groups was performed using the Kruskal-Wallis test. Post hoc comparisons were made using the Bonferroni correction for multiple comparisons. We assessed the missing data and excluded variables with missing data exceeding 30% from the analysis, such as grandiose delusions, irritability, and affective ambivalence. The tolerance and the variance inflation factor (VIF) were calculated for each independent variable of the final model to detect multicollinearity. VIFs over 10 indicated significant multicollinearity that needed to be corrected (*Menard,*

*2002*). Binary logistic regressions (forward: LR) were conducted, variables with $P < 0.10$ in the univariate analysis used as independent variables and violent crime used as the dependent variable, to examine the independent correlates of violent crime. The level of statistical significance was set at 0.05 (two-tailed).

## RESULTS

### Sociodemographic characteristics

The patient recruitment process is presented in Fig. 1. A total of 447 (94.5%) patients completed the interview. In the violent group, six were excluded for hearing difficulties, 10 were excluded for uncooperativeness and five were excluded for incomplete information. In the non-violent control group, three were excluded due to hearing difficulties and two were excluded for uncooperativeness. Finally, 308 patients with schizophrenia who engaged in violent criminal cases (155 in the homicide group and 153 in the intentional injury group) and 139 non-violent patients with schizophrenia were included in this study. The median and IQR for both violent and non-violent groups were 34.0 (28.0–42.0). There were no significant differences between the two groups in age, ethnicity, marital status, and family history of mental disorders. Compared to the non-violent group, the violent group was more likely to be males ($P = 0.020$), have low educational attainment ($P = 0.042$), be unemployed ($P = 0.024$), and live in rural areas ($P = 0.002$). Furthermore, the homicide group was more likely to live in rural areas compared to the non-violent group (see Table 1).

### Clinical characteristics

The violent group did not differ from the non-violent group in terms of duration of illness, alcohol abuse, history of drug use, and history of self-harm. However, there were significantly fewer patients having received previous treatment ($P < 0.001$), regular treatment in the previous four weeks ($P < 0.001$), but more patients having a history of violence ($P < 0.001$) in the violent group.

Compared to the intentional injury group, we found that the homicide group was less likely to have received a regular treatment in the previous four weeks. Additionally, compared to the non-violent group, the homicide group exhibited lower probabilities of previous treatment, regular treatment in the previous four weeks. They were more likely to have a history of violence (Table 2).

### Psychopathology and violence

There was no difference between the violent group and the non-violent group with regard to command hallucinations, depression, and excitement. Compared to the non-violent group, the violent group were more likely to have persecutory delusions ($P < 0.001$), delusions of jealousy ($P = 0.004$), and apathy ($P = 0.011$). Additionally, they exhibit a longer duration of untreated psychosis ($P = 0.028$) and poorer insight ($P < 0.001$). Furthermore, in comparison to the non-violent group, the homicide group were more likely to have persecutory delusions and apathy, along with poorer insight (see Table 2).

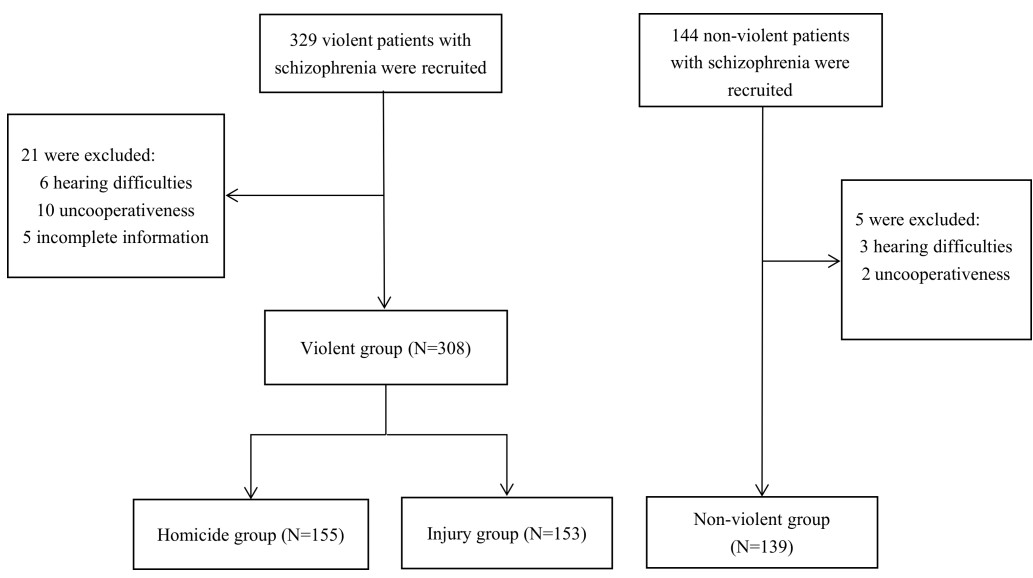

**Figure 1** Flow chart of the recruitment process.

**Table 1** The demographic data of the groups.

| Variable | A: Homicide (N = 155) | B: Intentional injury (N = 153) | C: Violent (N = 308) | D: Non-violent (N = 139) | P (C vs. D) | P (A vs. B vs. D) | Post Hoc analysis |
|---|---|---|---|---|---|---|---|
| Age (years) | 35.0 (27.0–43.0) | 34.0 (29.0–42.0) | 34.0 (28.0–42.0) | 34.0 (28.0–42.0) | 0.814[a] | 0.883[b] | |
| Male | 129 (83.2%) | 130 (85.0%) | 259 (84.1%) | 104 (74.8%) | **0.020**[c] | 0.062[c] | |
| Junior middle school and below | 109 (70.3%) | 103 (67.3%) | 212 (68.8%) | 82 (59.0%) | **0.042**[c] | 0.109[c] | |
| Unemployed | 142 (91.6%) | 141 (92.2%) | 283 (91.9%) | 118 (84.9%) | **0.024**[c] | 0.078[c] | |
| Living in rural areas | 137 (88.4%) | 123 (80.4%) | 260 (84.4%) | 100 (71.9%) | **0.002**[c] | **0.002**[c] | D <A |
| Han Chinese | 139 (89.7%) | 142 (92.8%) | 281 (91.2%) | 132 (95.0%) | 0.168[c] | 0.226[c] | |
| Unmarried | 106 (68.4%) | 109 (71.2%) | 215 (69.8%) | 84 (60.4%) | 0.051[c] | 0.130[c] | |
| Family history of mental disorders | 11 (7.1%) | 9 (5.9%) | 20 (6.5%) | 7 (5.0%) | 0.549[c] | 0.756[c] | |

**Notes.**
Bold indicates significant findings.
[a]Mann Whitney *U* test.
[b]Kruskal–Wallis test.
[c]Chi-square test.
[d]Fisher's exact test.

## Risk model for violence

The variables that demonstrated significance in the univariate analysis, including gender, education, work status, household registration, previous treatment, duration of untreated psychosis, regular treatment in the previous four weeks, history of violence, persecutory delusions, delusions of jealousy, apathy, and insight, incorporated into a binary logistic regression model. This model aimed to ascertain the predictive factors associated with violent behaviors. After gradually adjusting for all potential confounders, persecutory

**Table 2  The psychiatric data of the groups.**

| Variable | A:Homicide (N = 155) | B: Intentional injury (N = 153) | C: Violent (N = 308) | D: Non-violent (N = 139) | P (C vs. D) | P (A vs. B vs. D) | Post Hoc analysis |
|---|---|---|---|---|---|---|---|
| Duration of illness (years) | 7.0 (3.0–12.0) | 7.0 (2.0–12.5) | 7.0 (3.0–12.0) | 7.0 (4.0–14.0) | 0.142[a] | 0.270[b] | |
| Previous treatment | 107 (69.0%) | 101 (66.0%) | 208 (67.5%) | 118 (84.9%) | **<0.001**[c] | **<0.001**[c] | A <D, B <D |
| Regular treatment in the previous four weeks | 10 (6.5%) | 24 (15.7%) | 34 (11.0%) | 49 (35.3%) | **<0.001**[c] | **<0.001**[c] | A <B, A <D, B <D |
| History of violence | 88 (56.8%) | 78 (51.0%) | 166 (53.9%) | 39 (28.1%) | **<0.001**[c] | **<0.001**[c] | A >D, B >D |
| Alcohol abuse | 3 (1.9%) | 0 | 3 (1.0%) | 1 (0.7%) | 1.000[d] | 0.221[d] | |
| History of drug use | 2 (1.3%) | 0 | 2 (0.6%) | 1 (0.7%) | 1.000[d] | 0.535[d] | |
| History of self-harm | 17 (11.0%) | 10 (6.5%) | 27 (8.8%) | 10 (7.2%) | 0.577[c] | 0.316[c] | |
| Duration of un-treated psychosis (years) | 2.0 (0.0–6.0) | 2.0 (0.0–8.0) | 2.0 (0.0–7.0) | 1.0 (0.0–6.0) | 0.028[a] | 0.083[b] | |
| Command hallucinations | 13 (8.4%) | 13 (8.5%) | 26 (8.4%) | 8 (5.8%) | 0.321[c] | 0.611[c] | |
| Persecutory delusions | 102 (65.8%) | 89 (58.2%) | 191 (62.0%) | 49 (35.3%) | **<0.001**[c] | **<0.001**[c] | A >D, B >D |
| Delusions of jealousy | 8 (5.2%) | 10 (6.5%) | 18 (5.8%) | 0 | **0.004**[c] | **0.012**[c] | B >D |
| Apathy | 45 (29.0%) | 27 (17.6%) | 72 (23.4%) | 18(12.9%) | **0.011**[c] | **0.002**[c] | A >D |
| Insight | 3 (1.9%) | 10 (6.5%) | 13 (4.2%) | 35 (25.2%) | **<0.001**[c] | **<0.001**[c] | A <D, B <D |
| Depression | 8 (5.2%) | 14 (9.2%) | 22 (7.1%) | 8 (5.8%) | 0.587[c] | 0.324[c] | |
| Excitement | 7 (4.5%) | 3 (2.0%) | 10 (3.2%) | 2 (1.4%) | 0.357[d] | 0.210[d] | |

**Notes.**

Bold indicates significant findings.

[a] Mann Whitney $U$ test.

[b] Kruskal–Wallis test.

[c] Chi-square test.

[d] Fisher's exact test.

delusions, history of violence, regular treatment in the previous four weeks, and insight were independent predictors of violent behaviors. The results of the multicollinearity test indicate that the tolerance values for the variables included in the final model are all greater than 0.2, and all variance inflation factors (VIFs) are less than 2 (see Table 3). This indicates that there is no covariance. Later, these independent predictors were used as independent variables and violence as dependent variables to construct the risk model. Generally, factors with OR values greater than 1 were considered as risk factors for violence, whereas those with OR values less than 1 were regarded as protective factors for violence. Specifically, the risk model of violence can be presented as the following equation:

$$\text{Logit} P = 0.30 - 1.25 * \text{regular treatment in the previous four weeks} - 1.22 * \text{insight} + 1.06 * \text{violence history} + 0.95 * \text{delusions of persecutory}.$$

**Table 3 Multicollinearity test between independent variables of the final model.**

| Variables | Tolerance | VIF |
|---|---|---|
| Persecutory delusions | 0.95 | 1.05 |
| History of violence | 0.96 | 1.04 |
| Regular treatment in the previous four weeks | 0.91 | 1.10 |
| Insight | 0.85 | 1.18 |

**Notes.**
VIF, variance inflation factor.

**Table 4 Regression analysis of the predictive value of factors for violent crimes in schizophrenia.**

| Variable | B | SE | Wald | df | Odds ratio | 95% CI | P |
|---|---|---|---|---|---|---|---|
| History of violence | 1.06 | 0.24 | 19.07 | 1 | 2.88 | 1.79–4.64 | <0.001 |
| Persecutory delusions | 0.95 | 0.23 | 16.42 | 1 | 2.57 | 1.63–4.06 | <0.001 |
| Regular treatment in the previous four weeks | −1.25 | 0.29 | 18.70 | 1 | 0.29 | 0.16–0.51 | <0.001 |
| Insight | −1.22 | 0.38 | 10.25 | 1 | 0.30 | 0.14–0.62 | 0.001 |

Hosmer-Lemeshow test showed the fitness of this risk model was acceptable ($\chi^2 = 1.044$, $P = 0.959$); the total degree of accuracy of this model was 75.8%, and the degree of accuracy for the absence and presence of violence was 35.3% and 94.2%, respectively, indicating that this model was sensitive for the identification of the risk of violence. After controlling for sociodemographic variables (*e.g.*, gender, age). The multivariate analysis revealed that persecutory delusions (OR = 2.57, 95% CI [1.63–4.06]) and history of violence (OR = 2.88, 95% CI [1.79–4.64]) were independently associated with increased risk of violent behaviors, whereas regular treatment in the previous four weeks (OR = 0.29, 95% CI [0.16–0.51]) and insight (OR = 0.30, 95% CI [0.14–0.62]) were independently associated with reduced risk of violent behaviors (see Table 4).

## DISCUSSION

This study found that a history of violence and persecutory delusions were associated with an elevated risk of violence, and that regular treatment in the previous four weeks and insight were associated with a decreased risk of violence. These findings may help identify patients with schizophrenia who are at a higher risk for violence.

Previous studies have argued that dynamic risk factors are factors with changeable nature or severity, and fluctuations in their nature or severity are associated with changes in the risk of violence (*Hanson & Harris, 2001*) In contrast, static risk factors are those that are not amendable or those that change very little or very slowly over time (*Andrews & Bonta, 2010*). Our results suggested that a history of violence and no regular treatment in the previous four weeks were static risk factors for violence, while persecutory delusions and poor insight were dynamic risk factors for violence.

In the present study, a history of violence was found to be a risk factor for violence in the patients with schizophrenia. A meta-analysis showed that prior violent behaviors are a significant predictor of violent crimes in patients with schizophrenia (*Witt, Lichtenstein & Fazel, 2015*; *Fazel et al., 2017*), especially for homicide crimes (*Ntounas*

*et al., 2018*). Furthermore, a study found that prior violent behaviors remained a good predictor of violence in patients with schizophrenia even after controlling for the effect of psychopathological factors (*Hachtel et al., 2018*). Thus, the history of violent behaviors is highly likely to be an independent risk factor for recurrence of violence in patients with schizophrenia.

Our findings strongly indicated that persecutory delusions were correlated with an increased risk of violence. This observation aligns with numerous studies that have reported a significant association between persecutory delusions and heightened tendencies towards violent behaviors, and even criminal acts, in patients with schizophrenia (*Ullrich, Keers & Coid, 2014*; *Howard, Hepburn & Khalifa, 2015*; *Coid et al., 2018*). Additionally, studies have suggested that higher rates of persecutory delusions in patients with schizophrenia are associated with a negative attributional style. For example, they are more likely to attribute their negative experiences to external factors (*So, Tang & Leung, 2015*) and tend to over-interpret unsafe or threatening factors in their environment (*Lim, Gleeson & Jackson, 2011*). These tendencies may also contribute to an elevated risk of violence.

The current study observed that prior treatment, particularly within the previous four weeks, emerged as a mitigating factor against the occurrence of violent crimes. This finding is substantiated by parallel studies. For instance, a study reported a notable 45% reduction in incidents of violent crimes among patients who were receiving antipsychotic medications or mood stabilizers, in comparison to those who did not receive medication during the equivalent timeframe (*Fazel et al., 2014b*). Furthermore, within the cohort of 74,925 individuals documented in the Swedish National Register who were prescribed antipsychotic medications between 2006 and 2013, those under antipsychotic treatment exhibited a significantly lower relative risk across all categories of criminal outcomes (*Sariaslan et al., 2022*). Additionally, the study revealed that antipsychotic therapy exerted a notable risk-reducing effect on violent behaviors linked with psychopathological factors in patients diagnosed with schizophrenia (*Swanson et al., 2008*). Some studies have suggested that individuals receiving treatment for their mental disorders may experience a diminished risk of violence, potentially equivalent to that of the general population (*Langeveld et al., 2014*), and that even irregular treatment may still yield a reduction in the risk of violence, albeit to a lesser extent (*Keers et al., 2014*). Collectively, these findings underscore the pivotal role of pharmacological intervention in both disease management and the mitigation of violence risk in patients with schizophrenia.

The present study found that insight was a protective factor for violent crimes in patients with schizophrenia. It is generally accepted that poor insight is one of the key symptoms of schizophrenia and is associated with the severity of illness (*Mintz, Dobson & Romney, 2003*) as well as an increased risk of violent behaviors (*Arango et al., 1999*; *Buckley et al., 2004*). A meta-analysis found that poor insight was a greater contributor than positive symptoms to the risk of violence (*Witt, Van Dorn & Fazel, 2013*). However, some researchers have a different view on the relationship between insight and violence. A study found that patients with schizophrenia who had higher rates of insight were more likely to experience depression or negative emotions after learning of their actual conditions, depression or stigma; this might trigger their aggressive or hostile tendencies, leading to a higher risk

of violence (*Schandrin et al., 2019*). However, the disparities might have resulted from the difference in the definitions of insight.

The association between the history of substance abuse and violence were not found in the present study, this may be due to the low rates of alcohol abuse or drug use in the study population. Although many studies have identified comorbid alcohol and drug substance abuse as a prominent contributor to the risk of violent behaviors in patients with schizophrenia (*Räsänen et al., 1998*; *Fazel et al., 2009b*; *Hodgins, 2009*; *Langeveld et al., 2014*), the extremely low rates of alcohol abuse or drug use in China had precluded us from identifying such an association (*Thirthalli, Kumar & Arunachal, 2012*). Moreover, the measurement bias may have underestimated the impact of the history of comorbid substance abuse on violence, as individuals with both psychotic symptoms and psychoactive substance abuse are often diagnosed with a psychotic disorder instead of schizophrenia with substance abuse (*Liu et al., 2017*).

In summary, on the one hand, we need to identify and intervene early in patients with a history of violence to ensure that they have regular access to treatment and to reduce the increased risk of violence due to interruptions in treatment; on the other hand, we need to continuously and dynamically monitor and assess patients who exhibit delusions of persecution and a lack of insight in order to identify potential risks of violence in a timely manner. Particularly, it should be noted that, compared to the stable and unchanging sociodemographic and historical factors, the relatively dynamic and changeable treatment factors and psychiatric factors may have a greater significance in the assessment of violent risk.

## Limitations

Despite the strengths of this study, there are also some limitations. Firstly, as this study adopted a cross-sectional design, the temporal relationships between violent behaviors and its correlates could not be investigated in this study. As the relationship between psychopathological factors and violence might be dynamic and fluctuating over time (*Coid et al., 2013*; *Dorn et al., 2017*), longitudinal studies are warranted to examine such a relationship. Secondly, because the scope of this study is limited to Hunan Province, our results may not be universally applicable. Therefore, these findings should be interpreted and generalized with caution. Thirdly, this study relied on medical records and assessment notes to identify the presence of psychotic symptoms in patients before committing a crime, which may lead to the omission of symptoms and introduce certain biases. Additionally, because this study only assessed the presence or absence of symptoms, it is difficult to compare these symptoms with other studies or groups, thereby limiting the comparability of the research. We recommend that future studies conduct prospective studies, use standardized tools to assess psychotic symptoms, and establish more comprehensive recording systems to reduce these biases.

## CONCLUSIONS

Our study identified both static risk factors, such as a history of violence and lack of regular treatment in the previous four weeks, as well as dynamic risk factors, including

persecutory delusions and poor insight, associated with violent behavior in patients with schizophrenia. Given that violent crimes and homicides are serious consequences associated with schizophrenia, the identification of risk factors for violence holds significant potential for informing the development of more effective strategies to manage aggression in this patient population.

## ACKNOWLEDGEMENTS

The authors acknowledge the contribution of all the participants and collaborators of this study.

### Funding
This study was funded by the National Natural Science Foundation of China (82171509), STI2030-Major Projects (2021ZD0200700). The funders had no role in study design, data collection and analysis, decision to publish, or preparation of the manuscript.

### Grant Disclosures
The following grant information was disclosed by the authors:
the National Natural Science Foundation of China: 82171509.
STI2030-Major Projects: 2021ZD0200700.

### Competing Interests
The authors declare there are no competing interests.

### Author Contributions
- Ruoheng Lin conceived and designed the experiments, performed the experiments, analyzed the data, prepared figures and/or tables, authored or reviewed drafts of the article, and approved the final draft.
- Qiguang Li conceived and designed the experiments, performed the experiments, analyzed the data, prepared figures and/or tables, authored or reviewed drafts of the article, and approved the final draft.
- Ziwei Liu conceived and designed the experiments, performed the experiments, analyzed the data, authored or reviewed drafts of the article, and approved the final draft.
- Shaoling Zhong conceived and designed the experiments, authored or reviewed drafts of the article, and approved the final draft.
- Ying Huang conceived and designed the experiments, performed the experiments, analyzed the data, authored or reviewed drafts of the article, and approved the final draft.
- Hui Cao conceived and designed the experiments, performed the experiments, authored or reviewed drafts of the article, and approved the final draft.
- Xiangbin Zhang conceived and designed the experiments, authored or reviewed drafts of the article, and approved the final draft.

- Jiansong Zhou conceived and designed the experiments, prepared figures and/or tables, and approved the final draft.
- Xiaoping Wang conceived and designed the experiments, prepared figures and/or tables, authored or reviewed drafts of the article, and approved the final draft.

## Human Ethics

The following information was supplied relating to ethical approvals (*i.e.*, approving body and any reference numbers):

The Ethics Committee of the Second Xiangya Hospital.

## Data Availability

The raw measurements are available in the Supplementary File.

## Supplemental Information

Supplemental information for this article can be found online at http://dx.doi.org/10.7717/peerj.18014#supplemental-information.

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
