# Peer review of "Risk factors for violent crime in patients with schizophrenia: a retrospective study"

_PeerJ, doi:10.7717/peerj.18014_

## Round 0.1 · original submission · Major Revisions

I have now received the reviewers' comments on your manuscript. They have suggested some revisions to your manuscript. Therefore, I invite you to respond to the reviewers' comments and revise your manuscript.

Reviewer 1 ·

Basic reporting

The authors stated that China has a different situation from Western countries because China is a developing country. After this, the authors wrote, “But as the prevalence of substance abuse is much lower in both the general population (Hao et al., 2002) and the population with schizophrenia (Thirthalli, Kumar & Arunachal, 2012) in China,” This might have been true. But the citations were 12 and 22 years old. The country dramatically changed in these 22 years. What is the difference in substance abuse in the latest research between China and Western countries? Is the difference significant enough to justify this study?

Experimental design

As the authors wrote, the aim of the current study is “exploring the relationship between factors based on the Chinese population characteristics and violent crime among patients with schizophrenia, construct a model of violence risk assessment, and provide an accurate approach to violence prevention among Chinese patients with schizophrenia.” Indeed, the aim of this study is not new if it were conducted outside of China. Then, this paper should be published in a more local journal.

Why did the authors investigate such old data?
The threshold for statistical significance should be corrected for multiple comparisons.

Validity of the findings

The authors compared demographic and psychopathological characteristics between violent- and non-violent-groups. Are there any other analyses corroborating that the characteristics the authors identified were associated with violence? The results of the categorical comparison are too simple.

Additional comments

The authors wrote, “Since psychotic symptoms (especially delusions) may be culturally relevant and differ across cultures, for example, Zheng (Zheng, 1989) found significant differences between China and Japan in the content of exaggerated delusions, descent delusion, victimization delusions, and possession delusions, the relationship between psychotic symptoms and violence in Chinese patients with schizophrenia.” How about Japan?

Reviewer 2 ·

Basic reporting

Dear authors
In order to improve the quality of the manuscript, I offer a number of comments that I hope will be of interest to you:
-Abstract: Key words have been omitted. It is better to choose keywords based on mesh.
Introduction: A review of the existing literature has been made and the discussion in this field is discussed. In addition, the necessity of conducting a better study should be explained.
Methods: The reason for choosing this number of sample size should be explained and the sampling method should be expressed in a better way.
Findings: The data presented in the table do not need to be repeated in the text of the manuscript. Figure number one is not very clear and legible.
- Discussion: The key findings of the study should be discussed and compared with similar studies. In addition, the clinical and practical applications of the study should be stated.
- References: some references are written based on the format of the journal.

With respect

Experimental design

.

Validity of the findings

.

·

Basic reporting

The paper is written clearly in all its segments. The structure of the paper is in accordance with PeerJ standards, including the Abstract of the paper.
The introduction includes a literature review and a clearly defined research objective.
For this research, total of 447 patients with schizophrenia were included in this study, divided in two groups, first (308 patients) who engaged in violent criminal cases and second group of 139 non-violent patients with schizophrenia.
The discussion is correct. This study found that History of violence and Persecutory delusions were associated with an increased risk of violence, and that Treatment in the past four weeks and insight were associated with a decreased risk of violence.
Finally, in the Conclusion, the authors make suggestions for further research into the problem of identification of patients with schizophrenia who are at a higher risk of committing violence.
The literature contains 38 references, of which only one third of the references are from 5 years ago.

Experimental design

The design of the study is within the concept of the journal, while respecting high technical and ethical standards. Parts of the paper are transparent and logically organized, including one figure and four tables.
The method of investigation is informative and clear. The literature is clearly and adequately cited.

Validity of the findings

The obtained results can help in the identification of patients with schizophrenia who are at a higher risk of committing violence, emphasizing the key role of regular application of psychopharmacological therapy in the control of the disease with the aim of mitigating the risk of violence in patients with schizophrenia.
Important investigation of forensic psychiatry.
I recommend this paper to be published.

Additional comments

Important investigation of forensic psychiatry.

Reviewer 4 ·

Basic reporting

This study aimed to examine the risk factors for violent crime among schizophrenia patients through a retrospective case-control analysis. The authors compared 308 violent offenders with schizophrenia to 139 non-violent counterparts. They found that a history of violence, persecutory delusions, and lack of treatment or insight were significantly associated with violent behavior. This study is clinically relevant to help clinicians identify at-risk patients and formulate strategies to prevent violence in individuals with schizophrenia; however, some issues need to be addressed.
First, the authors described that they reviewed forensic files or assessment records and identified whether patients exhibited any psychotic symptoms within days to a week before the index offense. However, the reviewer is concerned that the absence of the description of a specific symptom in the records does not always mean the absence of the symptom. Please discuss this methodological limitation which would affect the results.
Second, while the study suggests practical implications for identifying and managing patients with schizophrenia at risk for violence, it could further elaborate on how these findings could be operationalized in clinical settings, including potential challenges and recommendations for practice.
Finally, the study acknowledges its cross-sectional design and the implications for causality assessment. However, discussing the potential impact of selection bias, the retrospective nature of data collection, and the study's geographical and cultural specificity on the findings would provide a more comprehensive view of the limitations.

Experimental design

Please see above.

Validity of the findings

Please see above.

Additional comments

None.

---

## Round 0.2 · accepted · Accept

I have carefully reviewed the manuscript, the reviewer comments, and the author responses, and I agree with reviewers 3 and 4 that comments were carefully and sufficiently addressed. the manuscript is ready for publication. Reviewer 1 comments that is is unclear how the study may or may not generalize to other East Asian countries, but the revised manuscript is careful to limit its conclusions to the population under study. As such, I am of the opinion that the manuscript is ready for publication.

Reviewer 1 ·

Basic reporting

I still don't think this article contains enough novelty or significance.

Experimental design

The data are old.

Validity of the findings

The authors wrote that China is the most populated country in the world, which is no longer valid. India is. The authors emphasized the significance by stating that the findings can be applied to other East Asian countries. However, South Korea and Japan have completely different social systems. If the authors said that the findings could be applicable in North Korea by using data from the northern part of China, the results would have much more significance because NK does not have data unlike SK or Japan.

·

Basic reporting

/

Experimental design

/

Validity of the findings

/

Additional comments

I have read the comments of the other three reviewers, and I agree with their suggestions for correcting the paper. Overall, I did not have any objections to the basic version of the paper, but I believe that the suggestions of the other reviewers have a positive effect on improving the quality of the final version of the paper.
So, I recommend this paper to be accepted for publication.

Reviewer 4 ·

Basic reporting

I have no additional comments.

Experimental design

I have no additional comments.

Validity of the findings

I have no additional comments.

Additional comments

I have no additional comments.